# IMAGE SUPER-RESOLUTION WITH
# TEXT PROMPT DIFFUSION

## ABSTRACT

Image super-resolution (SR) methods typically model degradation to improve reconstruction accuracy in complex and unknown degradation scenarios. However, extracting degradation information from low-resolution images is challenging, which limits the model performance. To boost image SR performance, one feasible approach is to introduce additional priors. Inspired by advancements in multimodal methods and text prompt image processing, we introduce text prompts to image SR to provide degradation priors. Specifically, we first design a text-image generation pipeline to integrate text into the SR dataset through the text degradation representation and degradation model. The text representation applies a discretization manner based on the binning method to describe the degradation abstractly. This method maintains the flexibility of the text and is user-friendly. Meanwhile, we propose the PromptSR to realize the text prompt SR. The PromptSR utilizes the pre-trained language model (*e.g.*, T5 or CLIP) to enhance restoration. We train the PromptSR on the generated text-image dataset. Extensive experiments indicate that introducing text prompts into SR, yields excellent results on both synthetic and real-world images. The code will be released.

## 1 INTRODUCTION

Single image super-resolution (SR) aims to recover high-resolution (HR) images from their corresponding low-resolution (LR) counterparts. Over recent years, the proliferation of deep learning-based methods (Dong et al., 2014; Zhang et al., 2018c; Chen et al., 2023) has significantly advanced this domain. Nevertheless, the majority of these methods are trained with known degradation (*e.g.*, bicubic interpolation), which limits their generalization capabilities (Wang et al., 2021; Zhang et al., 2023b). Consequently, these methods face challenges when applied to scenarios with complex and diverse degradations, such as real-world applications.

A feasible approach to tackle the diverse SR challenges is blind SR. Blind SR focuses on reconstructing LR images with complex and unknown degradation, making it suitable for a wide range of scenarios (Liu et al., 2022). Methods within this realm can roughly be divided into several categories. **(1)** Explicit methods (Zhang et al., 2018a) typically rely on predefined degradation models. They estimate degradation parameters (*e.g.*, blur kernel or noise) as conditional inputs to the SR model. However, the predefined degradation models exhibit a limited degradation representation scope, restricting the generality of methods. **(2)** Implicit methods (Cai et al., 2019; Wei et al., 2021) capture underlying degradation models through extensive external datasets. They achieve this by leveraging real-captured HR-LR image pairs, or HR and unpaired LR data, to learn the data distribution. Nevertheless, learning the data distribution is challenging, with unsatisfactory results. **(3)** Currently, another image SR paradigm (Zhang et al., 2021; Wang et al., 2021) is popularized: defining complex degradation to synthesize a large amount of data for training. To simulate real-world degradation, these approaches set the degradation distribution sufficiently extensive. Nonetheless, this increases the learning difficulty of the SR model and inevitably causes a performance drop.

In summary, the modeling of degradation is crucial to image SR, typically in complex application scenarios. However, most methods extract degradation information mainly from LR images, which is challenging and limits performance. One approach to advance SR performance is to introduce additional priors, such as reference priors (Jiang et al., 2021) or generative priors (Chan et al., 2021; Yang et al., 2021). **Motivated** by recent advancements in the multi-modal model (Radford et al.,

LR   HR   Bicubic   w/o text prompt   w/ text prompt

Figure 1: Visual comparison ($\times 4$). The LR image undergoes complex and unknown degradations (*e.g.*, blur, noise, and downsampling). By introducing text prompts (*e.g.*, [***heavy blur, upsample, medium noise, medium compression, downsample***], in the instance) into the SR task to provide degradation priors, the reconstruction quality can be effectively improved.

2021; Liu et al., 2023), text prompt image generation (Ramesh et al., 2021; Rombach et al., 2022), and manipulation (Brooks et al., 2023), we introduce the text prompt to provide priors for image SR. This approach offers several advantages: **(1)** Textual information is inherently flexible and suitable for various situations. **(2)** The power of the current pre-trained language model can be leveraged. **(3)** Text guidance can serve as a complement to current methods for image SR.

In this work, we propose a method to introduce text as additional priors to enhance image SR. Our design encompasses two aspects: the dataset and the model, with two motivations. **(1) Dataset:** For text prompt SR, large-scale multi-modal (text-image) data is crucial, yet challenging to collect manually. As mentioned above, the degradation models (Wang et al., 2021) can synthesize vast amounts of HR-LR image pairs. Hence, we consider incorporating text into the degradation model to generate the corresponding data. **(2) Model:** Text prompt SR inherently involves text processing. Meanwhile, the pre-trained language models possess powerful textual understanding capabilities. Thus, we utilize these models within our model to enhance text guidance and improve restoration.

Specifically, we develop a text-image generation pipeline that integrates text into the SR degradation model. **Text prompt for degradation:** We utilize text prompts to represent the degradation to provide additional prior. Since the LR image could provide the majority of low-frequency (Zhang et al., 2018c) and semantic information related to the content (Rombach et al., 2022), we care little about the abstract description of the overall image. **Text representation:** We first discretize degradation into components (*e.g.*, blur, noise). Then, we employ the binning method (Zhang et al., 2023b) to partition the degradation distribution, describe each segment textually, and merge them, to get the final text prompt. This discrete approach simplifies representation, which is intuitive and user-friendly to apply. **Flexible format:** To enhance prompt practicality, we adopt a more flexible format, such as arbitrary order or simplified (*e.g.*, only noise description) prompts. The recovery results, benefiting from the generalization of prompts, are also remarkable. Details are shown in Sec. 4.2.3. **Text-image dataset:** We adopt degradation models akin to previous methods (Zhang et al., 2021; Wang et al., 2021) to generate HR-LR image pairs. Simultaneously, we utilize the degradation description approach to produce the text prompts, thus generating the text-image dataset.

We further propose a network, PromptSR, to realize the text prompt image SR. Our PromptSR leverages the advanced diffusion model (Ho et al., 2020; Saharia et al., 2022b) for high-quality image restoration. Moreover, as analyzed previously, we apply the pre-trained language model (*e.g.*, T5 (Raffel et al., 2020) or CLIP (Radford et al., 2021)) to improve recovery. In detail, the language model acts as the text encoder to map the text prompt into a sequence of embeddings. The diffusion model then generates corresponding HR images, conditioned on LR images and text embeddings. Trained on the text-image dataset, our PromptSR performs excellently on both synthetic and real-world images. For real images, we leverage multi-modal large language models (MLLMs) (OpenAI, 2023; Ye et al., 2024) to generate professional image quality assessments as prompts. As illustrated in Fig. 1, when applying the text prompt, the model reconstructs a more realistic and clear image.

Overall, we summarize the main contributions as follows:

- We introduce text prompts as degradation priors to advance image SR. This explores the application of textual information in the SR task.

- We develop a text-image generation pipeline that integrates the user-friendly and flexible prompt into the SR dataset via text representation and degradation model.

- We propose a network, PromptSR, to realize the text prompt SR. The PromptSR utilizes the pre-trained language model to improve the restoration.

- Extensive experiments show that the introduction of text prompts into image SR leads to impressive results on both synthetic and real-world images.

## 2 RELATED WORK

### 2.1 IMAGE SUPER-RESOLUTION

Numerous deep networks (Zhang et al., 2018c; Chen et al., 2022b) have been proposed to advance the field of image SR since the pioneering work of SRCNN (Dong et al., 2014). Meanwhile, to enhance the applicability of SR methods in complex (*e.g.*, real-world) applications, blind SR methods have been introduced. To this end, researchers have explored various directions (Liu et al., 2022). First, explicit methods predict the degradation parameters (*e.g.*, blur kernel or noise) as the additional condition for SR networks (Gu et al., 2019; Bell-Kligler et al., 2019; Zhang et al., 2020). For instance, SRMD (Zhang et al., 2018a) takes the LR image with an estimated degradation map for SR reconstruction. Second, implicit methods learn underlying degradation models from external datasets (Bulat et al., 2018). These methods include supervised learning using paired HR-LR datasets, such as LP-KPN (Cai et al., 2019). Third, simulate real-world degradation with a complex degradation model and synthesize datasets for supervised training (Zhang et al., 2023b; Chen et al., 2022a). For example, Real-ESRGAN (Wang et al., 2021) introduces a high-order degradation, while BSRGAN (Zhang et al., 2021) proposes a random shuffling strategy. However, most methods still face challenges in degradation modeling, thus restricting SR performance.

### 2.2 DIFFUSION MODEL

The diffusion model (DM) has shown significant effectiveness in various synthetic tasks, including image (Ho et al., 2020; Song et al., 2020), video (Bar-Tal et al., 2022), audio (Kong et al., 2020), and text (Li et al., 2022b). Concurrently, DM has made notable advancements in image manipulation and restoration tasks, such as image editing (Avrahami et al., 2022), inpainting (Lugmayr et al., 2022), and deblurring (Whang et al., 2022). In the field of SR, exploration has also been undertaken. SR3 (Saharia et al., 2022b) conditions DM with LR images to constrain output space and generate HR results. Moreover, some methods, like DDRM (Kawar et al., 2022) and DDNM (Wang et al., 2023), apply degradation priors to guide the reverse process of pre-trained DM. However, these methods are primarily tailored for known degradations (*e.g.*, bicubic interpolation). Currently, some approaches (Wang et al., 2024; Lin et al., 2024) leverage pre-trained DM and fine-tune it on synthetic HR-LR datasets for real-world SR tasks. Nevertheless, these methods still mainly employ LR images, disregarding the utilization of other modalities (*e.g.*, text) to provide priors.

### 2.3 TEXT PROMPT IMAGE PROCESSING

This field, which includes image generation and image manipulation, is rapidly evolving. For generation, the large-scale text-to-image (T2I) models are successfully constructed using the diffusion model and CLIP (Radford et al., 2021), *e.g.*, Stable Diffusion (Rombach et al., 2022) and DALL-E-2 (Ramesh et al., 2022). Imagen (Saharia et al., 2022a) further demonstrates the effectiveness of large pre-trained language models, *i.e.*, T5 (Raffel et al., 2020), as text encoders. Moreover, some methods (Zhang et al., 2023a; Qin et al., 2023), like ControlNet (Zhang et al., 2023a), integrate more conditioning controls into text-to-image processes, enabling finer-grained generation.

For manipulation, numerous methods (Hertz et al., 2022; Kawar et al., 2023; Kim et al., 2022; Avrahami et al., 2022; Brooks et al., 2023) have been proposed. For instance, StyleCLIP (Patashnik et al., 2021) combines StyleGAN (Karras et al., 2019) and CLIP (Radford et al., 2021) to manipulate images using textual descriptions. Meanwhile, several methods are based on pre-trained T2I models, *e.g.*, Stable Diffusion. For example, Prompt-to-Prompt (Hertz et al., 2022) edits synthesis images by modifying text prompts. Imagic (Kawar et al., 2023) achieves manipulation of real images by fine-tuning models on given images. InstructPix2Pix (Brooks et al., 2023) employs editing instructions to modify images without requiring a description of image content. However, in image SR, the utilization of text prompts has seldom been explored.

## 3 METHOD

We introduce text prompts into image SR to enhance the reconstruction results. Our design encompasses two aspects: the dataset and the model. **(1) Dataset:** We propose a text-image generation pipeline integrating text prompts into the SR dataset. Leveraging the binning method, we apply the text to realize simplified representations of degradation, and combine it with a degradation model to generate data. **(2) Model:** We design the PromptSR for image SR conditioned on both text and image. The network is based on the diffusion model and the pre-trained language model.

Figure 2: Illustration of the text-image generation pipeline. (a) The pipeline comprises the degradation model (top) and the text representation (bottom). The degradation model comprises five steps, where "Comp" denotes the compression. The text representation describes each degradation operation in a discretized manner, *e.g.*, [***medium noise***] for noise operation. Except for the illustrated aligned prompt-degradation sequence, our pipeline supports more flexible degradation and prompt formats, *e.g.*, random order or simplified. (b) An example to display the dataset.

## 3.1 TEXT-IMAGE GENERATION PIPELINE

To realize effective training, and enhance model performance, a substantial amount of text-image data is required. Current methods (Cai et al., 2019; Wang et al., 2021) generate data for image SR by manual collection or through degradation synthesis. However, there is a lack of large-scale multi-modal text-image datasets for the SR task. To address this issue, we design the text-image generate pipeline to produce the datasets ($\mathbf{c}$, $[\mathbf{y}, \mathbf{x}]$), as illustrated in Fig. 2, where $\mathbf{c}$ is the text prompt describing degradation; $[\mathbf{y}, \mathbf{x}]$ denotes HR and LR images, respectively. The pipeline comprises two components: a **degradation model** that generates HR-LR image pairs and a **text representation module** that produces text prompts describing the degradation.

### 3.1.1 DEGRADATION MODEL

We aim to reconstruct HR images from LR images with complex and unknown degradation. To encompass the typical degradations while maintaining design simplicity, we develop the degradation model, as depicted in Fig. 2a. Note that while the degradation process in the illustration is applied sequentially, our degradation pipeline supports the more **flexible** format, *e.g.*, random degradation sequences and the omission of certain components. We describe each component in detail.

***Blur.*** We employ two kinds of blur: isotropic and anisotropic Gaussian blur. The blur is controlled by the kernel with two parameters: kernel width $\eta$ and standard deviation $\sigma$.

***Resize.*** We upsample/downsample images using two resize with scale factors $\gamma_1$ and $\gamma_2$, respectively. We employ area, bilinear, and bicubic interpolation. The two-step resizing can broaden the degradation range and enhance the generality of the model. We demonstrate it in Sec. 4.2.2.

***Noise.*** We apply Gaussian and Poisson noise, with noise levels controlled by $\varphi_1$ and $\varphi_2$, respectively. Meanwhile, noise is randomly applied in either RGB or gray format.

***Compression.*** We adopt JPEG compression, a widely used compression standard, for image compression. The quality factor $q$ controls the image compression quality.

Given an HR image $\mathbf{y}$, we determine the degradation by randomly selecting the degradation method (*e.g.*, Gaussian noise or Poisson noise), and sampling all parameters (*e.g.*, noise level $\varphi_1$) from the uniform distribution. Through the degradation process, we obtain the corresponding LR image $\mathbf{x}$. Compared to other degradation models (*e.g.*, high-order (Wang et al., 2021)), ours maintains flexibility and simplicity while covering broad scenarios.

### 3.1.2 TEXT PROMPT

After generating HR-LR image pairs through the degradation model, we further provide descriptions for each pair as text prompts. Consequently, we incorporate text prompts into the dataset. This process encompasses two key considerations: **(1)** The specific content that should be described; **(2)** The user-friendly method for generating corresponding descriptions concisely and effectively. Given the characteristics of image SR, we utilize text to represent degradation. Meanwhile, we represent the degradation via a discretization manner based on the binning method (Zhang et al., 2023b).

***Text prompt for degradation.*** Typical text prompt image generation and manipulation methods (Ramesh et al., 2022; 2021; Avrahami et al., 2022) apply text prompts to describe the image content. These prompts often require semantic-level interpretation and processing of the image content. However, for the image SR task, it is crucial to prioritize fidelity to the original image. Meanwhile, LR images could provide the majority of the low-frequency information (Zhang et al., 2018c) and semantic information related to the content (Rombach et al., 2022). As shown in Fig. 3, elements like '*building*' and '*shutters*' in the caption prompt can be obtained from the LR image.

Therefore, we adopt the prompt for degradation, instead of the description of the overall image. This prompt can provide degradation priors and thus enhance the capability of methods to model degradation, which is crucial for image SR. As shown in Fig. 3, utilizing text to depict degradation, instead of the overall image content (Caption), yields restoration that is more aligned with the ground truth. To further demonstrate the effectiveness of text prompts for degradation, we provide more analyses in Sec. 4.2.4.

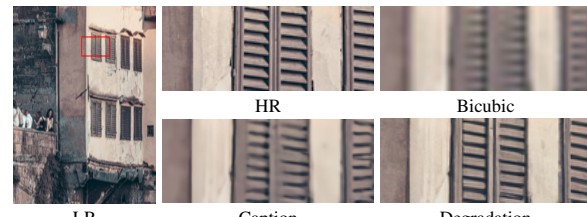

Figure 3: Visual comparison (×4) of different text contents. Caption (description of the overall image): [***people on a weathered balcony of a building with closed shutters***]. Degradation (ours): [***light blur, upsample, light noise, heavy compression, downsample***].

***Text representation.*** To facilitate data generation and practical usability, we describe degradation in natural language with the approach illustrated in Fig. 2a. Overall, we describe each degradation component via a discretized binning method, and combine them in a flexible format.

First, we discretize the degradation model into several components (*e.g.*, blur) and describe each using qualitative language via a binning method. The sampling distribution of parameters corresponding to each component is evenly divided into discrete intervals (bins). Each bin is summarized to represent the degradation. For instance, we divide the distribution of noise level $\varphi_1$ $[0, 9]$ into three uniform intervals ($[0, 3)$, $[3, 6)$, and $[6, 9]$), and describe them as '*light*', '*medium*', and '*heavy*'. Both Gaussian and Poisson noises are summarized as '*noise*', leading to the final representation: [***medium noise***]. Compared to specifying degradation names and their parameters, *e.g.*, [***Gaussian noise with noise level 4.5***], our discretized representation is more intuitive and **user-friendly**.

Finally, the overall degradation representation combines all component descriptions, *i.e.*, [***deblur description, ..., resize description***]. Figure 2b illustrates an example. The content of the prompt directly corresponds to the degradation. Furthermore, it is notable that, in our method, the prompt exhibits good generalization and supports **flexible** description formats. For instance, both arbitrary order or simplified (*e.g.*, only noise description) prompts can still lead to satisfactory restoration outcomes. In Sec. 4.2.3, we conduct a detailed investigation of the prompt format.

***Real-world application.*** For real-world images, users can utilize the latest multi-modal large language models (MLLMs) (Liu et al., 2023; OpenAI, 2023; Ye et al., 2024; Wu et al., 2024) to generate professional image quality assessments as prompts. This approach simplifies prompt generation for users. It also provides a pathway for improving image SR using MLLMs. Furthermore, users can fine-tune the MLLM-generated prompts based on the restoration results to achieve more **personalized** enhancements. More details are provided in the supplementary material.

## 3.2 PROMPTSR

PromptSR is based on the **general** diffusion model (Ho et al., 2020), commonly utilized for high-quality image restoration (Saharia et al., 2022b; Lin et al., 2024). Meanwhile, given the powerful capabilities of pre-trained language models (Radford et al., 2021; Raffel et al., 2020), we integrate them into the model to enhance performance. The architecture of our method is delineated in Fig. 4.

For the diffusion model, to underscore the effectiveness of text prompts, we employ a general text-to-image (T2I) diffusion architecture, rather than a meticulously designed structure. Specifically, our method employs a denoising network (DN), operating through a $T$-step reverse process to generate high-resolution (HR) images from Gaussian noise. The DN applies the U-Net structure (Ronneberger et al., 2015). It predicts the noise conditioned on the LR image (upsampled to the target resolution via bicubic interpolation) and text prompt.

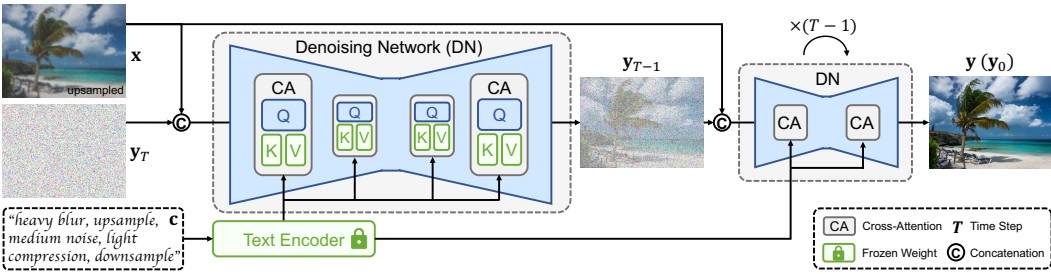

Figure 4: The overall architecture of the PromptSR. It comprises a denoising network (DN) and a pre-trained text encoder. The weights of the text encoder are frozen during training. The LR image **x** is first **upsampled** to the target HR image resolution via bicubic interpolation, then concatenated with the noise image $\mathbf{y}_t$ ($t \in [1, T]$) as input to the DN. The text prompt **c** is embedded by the text encoder. The embeddings are infused into the DN via the cross-attention (CA) module.

Concurrently, the pre-trained language model encodes the text prompts, where the information is integrated into feature maps of U-Net via the cross-attention module. By leveraging the powerful capabilities of the language model, our method can better understand degradation, thereby enhancing the restoration results. For more details on the PromptSR, please refer to the supplementary material.

### 3.2.1 PRE-TRAINED TEXT ENCODER

Text prompt image models (Patashnik et al., 2021; Avrahami et al., 2022; Rombach et al., 2022) mainly employ multi-modal embedding models, *e.g.*, CLIP (Radford et al., 2021), as text encoders. These encoders are capable of generating meaningful representations pertinent to tasks. Besides, compared to multi-modal embeddings, pre-trained language models (Devlin et al., 2019; Raffel et al., 2020) exhibit stronger text comprehension capabilities. Therefore, we attempt to apply different pre-trained text encoders to build a series of networks. These models demonstrate varying restoration performance levels, which we further analyze in Sec. 4.2.5.

### 3.2.2 TRAINING STRATEGY

We train the PromptSR using the text-image (**c**, [**y**, **x**]) dataset generated as described in Sec. 3.1. Given an HR image **y**, we add noise $\boldsymbol{\epsilon}$ through $t$ diffusion steps to obtain a noisy image $\mathbf{y}_t$, where $t$ is randomly sampled from $[1, T]$. The DN is conditioned on the LR image **x**, noisy image $\mathbf{y}_t$, and text prompt **c** to predict the added noise. The training objective is formulated as:

$$\mathcal{L} = \mathbb{E}_{\mathbf{y}, \mathbf{x}, \mathbf{c}, t, \boldsymbol{\epsilon} \sim \mathcal{N}(0,1)} \big[ |\boldsymbol{\epsilon} - \boldsymbol{\epsilon}_\theta(\mathbf{y}_t, \mathbf{x}, \boldsymbol{\tau}_\theta(\mathbf{c}), t)|_2^2 \big], \tag{1}$$

where $\boldsymbol{\epsilon}_\theta$ is the DN, while $\boldsymbol{\tau}_\theta$ is the text encoder. We freeze the weights of the text encoder and only train the DN. In this way, we can retain the original capabilities of the pre-trained model. Meanwhile, we can reduce training overhead by computing text embedding offline. After completing the training process, the PromptSR can be employed for both synthetic and real-world images. Benefiting from the multi-modal (text and image) design, it demonstrates excellent performance.

## 4 EXPERIMENTS

### 4.1 EXPERIMENTAL SETTINGS

#### 4.1.1 DEGRADATION SETTINGS

The degradation model in our proposed pipeline encompasses four operations: blur, resize, noise, and compression. Following previous methods (Wang et al., 2021; Zhang et al., 2021), the parameters for these operations are sampled from the uniform distribution. **Blur:** We adopt isotropic Gaussian blur and anisotropic Gaussian blur with equal probability. The kernel width $\eta$ is randomly selected from the set $\{7, 9, \ldots, 21\}$. The standard deviation $\sigma$ is sampled from a uniform distribution $\mathcal{U}_{[0.2,3]}$. **Resize:** We employ area, bilinear, and bicubic interpolation with probabilities of $[0.3, 0.4, 0.3]$. To expand the scope of degradation, we perform two resize operations at different stages. The first resize spans upsample and downsample, where the scale factor is $\gamma_1 \sim \mathcal{U}_{[0.15,1.5]}$. The second resize operation scales the resolution to $\frac{1}{4}$ of the HR image. **Noise:** We apply Gaussian and Poisson noise with equal probability. The level of Gaussian noise is $\varphi_1 \sim \mathcal{U}_{[1,30]}$, while the level of Poisson noise is $\varphi_2 \sim \mathcal{U}_{[0.05,3]}$. **Compression:** We employ JPEG compression with quality factor $q \sim \mathcal{U}_{[30,95]}$. Meanwhile, in all experiments, for simplifying implementation, unless expressly noted, the degradation and text prompt follow the fixed order and correspond one to one.

Table 1: Ablation study on the text prompt. We experiment on ControlNet (Zhang et al., 2023a) and PromptSR. We apply an empty string for without (✗) the prompt.

| Method | Text | LSDIR-Val | | DIV2K-Val | |
|---|---|---|---|---|---|
| | | LPIPS ↓ | DISTS ↓ | LPIPS ↓ | DISTS ↓ |
| ControlNet | ✗ | 0.3401 | 0.2059 | 0.3733 | 0.2396 |
| | ✓ | 0.3347 | 0.2054 | 0.3515 | 0.2306 |
| PromptSR | ✗ | 0.3473 | 0.2009 | 0.3384 | 0.1941 |
| | ✓ | 0.3211 | 0.1820 | 0.3086 | 0.1727 |

Table 2: Ablation study on the resizing operation. We compare the degradation with one resizing (keeping the first one) and two resizings.

| Method | Metric | One Resizing | Two Resizings |
|---|---|---|---|
| LSDIR-Val | LPIPS ↓ | 0.3709 | 0.3211 |
| | DISTS ↓ | 0.2254 | 0.1820 |
| DIV2K-Val | LPIPS ↓ | 0.3570 | 0.3086 |
| | DISTS ↓ | 0.2162 | 0.1727 |

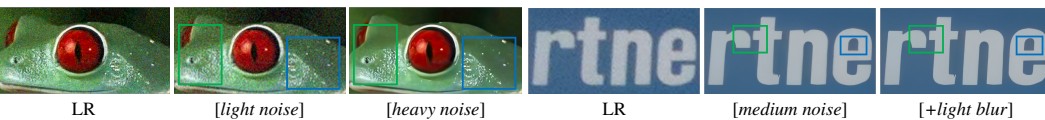

| LR | [*light noise*] | [*heavy noise*] | LR | [*medium noise*] | [*+light blur*] |

Figure 5: Visual results (×4) of different prompts. [...] represents the prompt. Boxes in the figures highlight the differences in details. Please zoom in for a better view.

### 4.1.2 DATASETS AND METRICS

We use the LSDIR (Li et al., 2023) as the training dataset. The LSDIR contains 84,991 high-resolution images. We generate the corresponding text-image dataset using our proposed pipeline. We evaluate our method on both synthetic and real-world datasets. For synthetic datasets, we employ Urban100 (Huang et al., 2015), Manga109 (Matsui et al., 2017), and the validation (Val) datasets of LSDIR and DIV2K (Timofte et al., 2017). For real-world datasets, we utilize RealSR (Cai et al., 2019). We also employ 45 real images directly captured from the internet, denoted as Real45. We conduct all experiments with a scale factor of ×4. To quantitatively evaluate our method, we adopt two traditional metrics: PSNR and SSIM (Wang et al., 2004), which are calculated on the Y channel of the YCbCr color space. We also utilize several perceptual metrics: LPIPS (Zhang et al., 2018b), ST-LPIPS (Ghildyal & Liu, 2022), DISTS (Ding et al., 2020), and CNNIQA (Kang et al., 2014). We further adopt an aesthetic metric: NIMA (Talebi & Milanfar, 2018).

### 4.1.3 IMPLEMENTATION DETAILS

The proposed PromptSR consists of two components: the denoising network (DN) and the pre-trained text encoder. The DN employs a U-Net architecture with a 4-level encoder-decoder. Each level contains two ResNet (He et al., 2016; Ho et al., 2020) blocks and one cross-attention block. For more detailed information about the DN model structure, please refer to the supplementary material. For the text encoder, we apply the pre-trained multi-modal model, CLIP (Radford et al., 2021). Additionally, we discuss other large language models, *e.g.*, T5 (Raffel et al., 2020), in Sec. 4.2.5.

We train our model on the generated text-image dataset with a batch size of 16 for a total of 1,000,000 iterations. The input image is randomly cropped to 64×64. We adopt the Adam optimizer (Kingma & Ba, 2015) with $\beta_1$=0.9 and $\beta_2$=0.99 to minimize the training objective (Eq. 1). The learning rate is $2\times10^{-4}$ and is reduced by half at the 500,000-iteration mark. For DM, we set the total time step $T$ as 2,000. For inference, we employ the DDIM sampling (Song et al., 2020) with 50 steps. We use PyTorch (Paszke et al., 2019) to implement our method with 4 Nvidia A100 GPUs.

### 4.2 ABLATION STUDY

We investigate the effects of our proposed method at SR (×4) task. We train all models on the LSDIR dataset with 500,000 iterations. We apply the validation datasets of LSDIR (Li et al., 2023) and DIV2K (Timofte et al., 2017) for testing. Results are shown in Fig. 5 and Tabs. 1, 2, 3, 4, and 5.

### 4.2.1 IMPACT OF TEXT PROMPT

We conduct an ablation to show the influence of introducing the text prompt into image SR. The results are listed in Tab. 1. To validate the effectiveness of the text prompts, rather than benefiting from the specialized network, we conduct experiments on ControlNet (Zhang et al., 2023a) and proposed PropmtSR. We take the LR image as the condition to ControlNet to realize SR. All four compared models are trained on LSDIR. For models that are without text prompts, we train and test using empty string. The comparison reveals that text prompts significantly enhance SR performance. It also demonstrates the universality of text prompts, applicable to various models.

Table 3: Ablation study on the format. (a) Random Order: shuffled degradation sequence. Fixed Order: fixed degradation sequence. (b) Random Order: mismatched prompt-degradation order. Simplified: randomly omitting 50% prompt contents. Original: aligned prompt-degradation order.

(a) Different degradation formats.

| Method | Random Order | | Fixed Order | |
|---|---|---|---|---|
| | LPIPS ↓ | DISTS ↓ | LPIPS ↓ | DISTS ↓ |
| LSDIR-Val | 0.3243 | 0.1860 | 0.3211 | 0.1820 |
| DIV2K-Val | 0.3193 | 0.1722 | 0.3086 | 0.1727 |

(b) Different prompt formats.

| Method | Random Order | | Simplified | | Original | |
|---|---|---|---|---|---|---|
| | LPIPS ↓ | DISTS ↓ | LPIPS ↓ | DISTS ↓ | LPIPS ↓ | DISTS ↓ |
| LSDIR-Val | 0.3231 | 0.1835 | 0.3268 | 0.1871 | 0.3211 | 0.1820 |
| DIV2K-Val | 0.3095 | 0.1730 | 0.3131 | 0.1767 | 0.3086 | 0.1727 |

Table 4: Ablation study on the text content. Caption: image content generated by BLIP (Li et al., 2022a). Degradation (ours): degradation process. Both: the combination of two.

| Method | LSDIR-Val | | DIV2K-Val | |
|---|---|---|---|---|
| | LPIPS ↓ | DISTS ↓ | LPIPS ↓ | DISTS ↓ |
| Caption | 0.3403 | 0.1931 | 0.3237 | 0.1840 |
| Degradation | 0.3211 | 0.1820 | 0.3086 | 0.1727 |
| Both | 0.3247 | 0.1884 | 0.3104 | 0.1770 |

Table 5: Ablation study on the pre-trained text encoder. We adopt different pre-trained language models as text encoders in our PromptSR. Params: the parameters of each text encoder.

| Method | Params | LSDIR-Val | | DIV2K-Val | |
|---|---|---|---|---|---|
| | | LPIPS ↓ | DISTS ↓ | LPIPS ↓ | DISTS ↓ |
| T5-small | 60M | 0.3260 | 0.1911 | 0.3218 | 0.1863 |
| CLIP | 428M | 0.3211 | 0.1820 | 0.3086 | 0.1727 |
| T5-xl | 3B | 0.3151 | 0.1753 | 0.3056 | 0.1682 |

Moreover, we visualize the impact of different prompts on the SR results in Fig. 5. We observe that the method can remove part of the noise for the image with severe noise when the prompt indicates [*light noise*] in the left instance. Conversely, a suitable prompt, *i.e.*, [*heavy noise*], can restore a more realistic result. Meanwhile, for images at the right, a simplified prompt, *i.e.*, [*medium noise*], can yield a relatively **satisfactory result**. Further refining the prompt, *i.e.*, [*+light blur*], can further improve the restoration outcome. These results demonstrate the flexibility of our prompts.

### 4.2.2 TWO RESIZING OPERATIONS

We investigate the different number of resizing operations in the degradation. The results are presented in Tab. 2. We can find that the model with two resizings performs better. This is because one single resizing is fixed at $\frac{1}{4}$ in the ×4 SR task. Introducing an additional resizing allows for variable scales, expands the degradation scope, and enhances the generality of the model.

### 4.2.3 FLEXIBLE FORMAT

We investigate the different formats of the degradation and prompt. The results are revealed in Tab. 3. Firstly, in Tab. 3a, we compare fixed and random degradation orders. The results indicate that random order slightly lowers performance. It may be because random order expands the degradation space (generalization), thus increasing training complexity and diminishing performance. To balance performance and generalization, we opt for the fixed order shown in Fig. 2.

Secondly, in Tab. 3b, we compare three prompt formats. The comparison shows that complete prompts (Original) reveal the best performance. Meanwhile, prompt order has little effect. Moreover, the simplified prompt can yield relatively good results due to the model generalization. Overall, our method exhibits fine generalization, supporting a flexible variety of degradation and prompt.

### 4.2.4 TEXT PROMPT FOR DEGRADATION

We study the effects of different content of text prompts. The results are presented in Tab. 4. We compare three types of text prompt content. All experiments are conducted on our proposed PromptSR. The comparison shows that descriptions of degradation (Degradation) are more suitable for the SR task than image content descriptions (Caption). This is consistent with our analysis in Sec. 3.1.2. Additionally, combining both descriptions results in a slight performance drop compared to using degradation prompts alone. This could be due to the disparity between the two descriptions, which hinders the utilization of degradation information provided by text prompts.

### 4.2.5 PRE-TRAINED TEXT ENCODER

We further explore the impact of different text encoders, with the results detailed in Tab. 5. We utilize several pre-trained text encoders: **CLIP** (Radford et al., 2021) (clip-vit-large) and **T5** (Raffel et al., 2020) (T5-small and T5-xl). We discover that models employing different text encoders display varied performance. Applying more powerful language models as text encoders enhances model performance. For instance, T5-xl, compared to T5-small, reduces the LPIPS on the LSDIR and DIV2K validation sets by 0.0109 and 0.0162, respectively. Moreover, it is also notable that the performance of the model is not entirely proportional to the parameter size of the text encoder. Considering both model performance and parameter size, we select CLIP as the text encoder.

Table 6: Quantitative comparison (×4) on synthetic datasets with state-of-the-art methods. The best and second-best results are colored red and blue.

| Dataset | Metric | DAN | Real-ESRGAN+ | BSRGAN | SwinIR-GAN | FeMaSR | Stable Diffusion | DiffBIR | PromptSR (ours) |
|---|---|---|---|---|---|---|---|---|---|
| Urban100 | PSNR ↑ | 21.12 | 20.89 | 21.66 | 20.91 | 20.37 | 20.201 | 21.73 | 21.39 |
| | SSIM ↑ | 0.5240 | 0.5997 | 0.6014 | 0.6013 | 0.5573 | 0.4852 | 0.5896 | 0.6130 |
| | LPIPS ↓ | 0.5835 | 0.2621 | 0.2835 | 0.2547 | 0.2725 | 0.4589 | 0.2586 | 0.2500 |
| | ST-LPIPS ↓ | 0.4457 | 0.2494 | 0.2748 | 0.2376 | 0.2442 | 0.3845 | 0.2686 | 0.2262 |
| | DISTS ↓ | 0.3125 | 0.1762 | 0.1857 | 0.1676 | 0.1877 | 0.2505 | 0.1857 | 0.1857 |
| | CNNIQA ↑ | 0.4033 | 0.6635 | 0.6247 | 0.6614 | 0.6781 | 0.5870 | 0.6517 | 0.6732 |
| | NIMA ↑ | 4.1485 | 5.3135 | 5.3671 | 5.3622 | 5.4161 | 4.6368 | 5.4010 | 5.5059 |
| Manga109 | PSNR ↑ | 21.78 | 21.62 | 22.26 | 21.81 | 21.46 | 18.76 | 21.37 | 20.82 |
| | SSIM ↑ | 0.6138 | 0.7217 | 0.7218 | 0.7258 | 0.6891 | 0.5412 | 0.6738 | 0.7048 |
| | LPIPS ↓ | 0.4238 | 0.2051 | 0.2194 | 0.2047 | 0.2145 | 0.3699 | 0.2198 | 0.1856 |
| | ST-LPIPS ↓ | 0.3396 | 0.1649 | 0.1789 | 0.1590 | 0.1520 | 0.2750 | 0.1679 | 0.1205 |
| | DISTS ↓ | 0.2101 | 0.1252 | 0.1396 | 0.1185 | 0.1418 | 0.1638 | 0.1380 | 0.1373 |
| | CNNIQA ↑ | 0.4172 | 0.6651 | 0.6550 | 0.6673 | 0.6735 | 0.6691 | 0.6988 | 0.6929 |
| | NIMA ↑ | 4.1478 | 4.9825 | 5.1913 | 4.8784 | 5.0625 | 4.6493 | 5.1738 | 5.4211 |
| LSDIR-Val | PSNR ↑ | 22.71 | 22.40 | 22.95 | 22.34 | 21.19 | 19.91 | 22.63 | 22.44 |
| | SSIM ↑ | 0.5578 | 0.6115 | 0.6067 | 0.6067 | 0.5542 | 0.4487 | 0.5725 | 0.6070 |
| | LPIPS ↓ | 0.6038 | 0.2932 | 0.3103 | 0.2911 | 0.2917 | 0.4489 | 0.3104 | 0.2810 |
| | ST-LPIPS ↓ | 0.4354 | 0.2502 | 0.2727 | 0.2440 | 0.2362 | 0.3521 | 0.2827 | 0.2258 |
| | DISTS ↓ | 0.2760 | 0.1627 | 0.1713 | 0.1598 | 0.1533 | 0.2240 | 0.1758 | 0.1548 |
| | CNNIQA ↑ | 0.3924 | 0.6417 | 0.5960 | 0.6277 | 0.6716 | 0.6563 | 0.5339 | 0.6726 |
| | NIMA ↑ | 4.0724 | 4.9878 | 5.0790 | 4.9551 | 5.1998 | 4.4452 | 5.1883 | 5.2538 |
| DIV2K-Val | PSNR ↑ | 24.98 | 25.24 | 25.73 | 25.73 | 23.80 | 21.47 | 25.56 | 25.14 |
| | SSIM ↑ | 0.6052 | 0.7017 | 0.6925 | 0.6932 | 0.6310 | 0.5120 | 0.6653 | 0.6813 |
| | LPIPS ↓ | 0.6315 | 0.2896 | 0.3006 | 0.2854 | 0.2899 | 0.4709 | 0.2973 | 0.2753 |
| | ST-LPIPS ↓ | 0.4487 | 0.2186 | 0.2259 | 0.2090 | 0.2061 | 0.2307 | 0.3717 | 0.1913 |
| | DISTS ↓ | 0.2668 | 0.1548 | 0.1632 | 0.1497 | 0.1451 | 0.2239 | 0.1809 | 0.1484 |
| | CNNIQA ↑ | 0.3897 | 0.6238 | 0.5908 | 0.6125 | 0.6617 | 0.5814 | 0.6380 | 0.6748 |
| | NIMA ↑ | 4.0737 | 4.8202 | 4.9330 | 4.8015 | 5.0451 | 4.3881 | 5.0213 | 5.0834 |

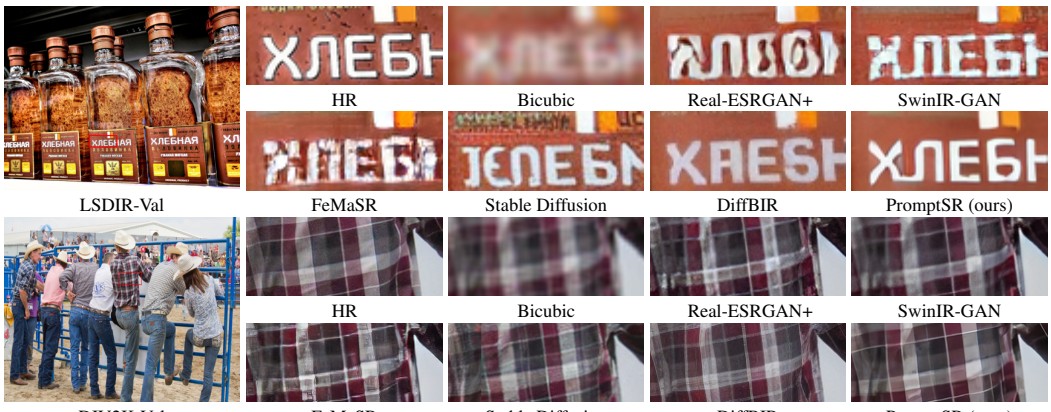

Figure 6: Visual comparison (×4) on synthetic datasets with state-of-the-art methods. Our method restores images with high realism and fidelity. Please zoom in for a better view.

## 4.3 EVALUATION ON SYNTHETIC DATASETS

We compare our method with several recent state-of-the-art methods: DAN (Huang et al., 2020), Real-ESRGAN+ (Wang et al., 2021), BSRGAN (Zhang et al., 2021), SwinIR-GAN (Liang et al., 2021), FeMaSR (Chen et al., 2022a), Stable Diffusion (Rombach et al., 2022), and DiffBIR (Lin et al., 2024). We show quantitative results in Tab. 6 and visual results in Fig. 6.

### 4.3.1 QUANTITATIVE RESULTS

We evaluate our method on some synthetic test datasets: Urban100 (Huang et al., 2015), Manga109 (Matsui et al., 2017), LSDIR-Val (Li et al., 2023), and DIV2K-Val (Timofte et al., 2017) in Tab. 6. Our method outperforms others on most **perceptual metrics**. For instance, compared to the suboptimal model SwinIR-GAN (Liang et al., 2021), our method reduces the LPIPS by 0.0101 on the DIV2K-Val dataset. Meanwhile, compared with DiffBIR (Lin et al., 2024), our PromptSR achieves a reduction in LPIPS by 0.0294 and 0.0220 on LSDIR-Val and DIV2K-Val, respectively. Moreover, for PSNR and SSIM, the two metrics are only used as references, since they do not consistently align well with the image quality (Saharia et al., 2022b). These quantitative results demonstrate that introducing text prompts into image SR can effectively improve performance.

Table 7: Quantitative comparison (×4) on the real-world dataset with state-of-the-art methods. The best and second-best results are colored red and blue.

| Dataset | Metric | DAN | Real-ESRGAN+ | BSRGAN | SwinIR-GAN | FeMaSR | Stable Diffusion | DiffBIR | PromptSR (ours) |
|---------|--------|-----|--------------|--------|------------|--------|------------------|---------|-----------------|
| RealSR | PSNR ↑ | 27.82 | 25.62 | 27.04 | 26.54 | 25.74 | 24.11 | 27.42 | 26.71 |
| | SSIM ↑ | 0.7978 | 0.7582 | 0.7911 | 0.7918 | 0.7643 | 0.6980 | 0.7790 | 0.7821 |
| | LPIPS ↓ | 0.4041 | 0.2843 | 0.2657 | 0.2765 | 0.2938 | 0.5035 | 0.3434 | 0.2702 |
| | ST-LPIPS ↓ | 0.3798 | 0.2165 | 0.1978 | 0.2078 | 0.1990 | 0.4122 | 0.2506 | 0.1937 |
| | DISTS ↓ | 0.2362 | 0.1732 | 0.1730 | 0.1672 | 0.1927 | 0.2441 | 0.2140 | 0.1820 |
| | CNNIQA ↑ | 0.2583 | 0.5755 | 0.5626 | 0.5208 | 0.5916 | 0.4465 | 0.5544 | 0.6376 |
| | NIMA ↑ | 3.9388 | 4.7673 | 4.8896 | 4.7338 | 4.8745 | 4.1598 | 4.8295 | 4.8917 |

Figure 7: Visual comparison (×4) on real-world datasets with state-of-the-art methods. Our method can generate more realistic images. Please zoom in for a better view.

### 4.3.2 VISUAL RESULTS

We show some visual comparisons in Fig. 6. We can observe that our proposed PromptSR is capable of restoring clearer and more realistic images, in some challenging cases. This is consistent with the quantitative results. Furthermore, we provide more visual results in the supplementary material.

## 4.4 EVALUATION ON REAL-WORLD DATASETS

We further evaluate our method on real-world datasets. We apply our PromptSR for real image SR by MLLM-generated prompts as depicted in Sec. 3.1.2. For instance, the prompt for the first case in Fig. 7: [*light blur, unchange, light noise, heavy compression, downsample*]. More prompts on real-world images are provided in the supplementary material.

### 4.4.1 QUANTITATIVE RESULTS

We present the quantitative comparison on RealSR (Cai et al., 2019) in Tab. 7. Our PromptSR achieves the best performance on most perceptual and aesthetic metrics, including ST-LPIPS, CN-NIQA, and NIMA. Meanwhile, it also scores well on LPIPS. These results further demonstrate the superiority of introducing text prompts into image SR tasks.

### 4.4.2 VISUAL RESULTS

We present some visual results in Fig. 7. Except for the RealSR dataset, we also conduct an evaluation on the Real45 dataset, collected from the internet. Our proposed method also outperforms other methods on real-world datasets. More comparison are provided in the supplementary material.

## 5 CONCLUSION

In this work, we introduce the text prompts to provide degradation priors for enhancing image SR. Specifically, we develop a text-image generation pipeline to integrate text into the SR dataset, via text degradation representation and degradation model. The text representation is flexible and user-friendly. Meanwhile, we propose the PromptSR to realize the text prompt SR. The PromptSR applies the pre-trained language model to enhance text guidance and improve performance. We train our PromptSR on the generated text-image dataset and evaluate it on both synthetic and real-world datasets. Extensive experiments demonstrate the effectiveness of introducing text into SR.

## ETHICS STATEMENT

The research conducted in the paper conforms, in every respect, with the ICLR Code of Ethics.

## REPRODUCIBILITY STATEMENT

We have provided implementation details in Sec. 4.1. We will also release all the code and models.

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
