# Supplementary Material: Image Super-Resolution with Text Prompt Diffusion

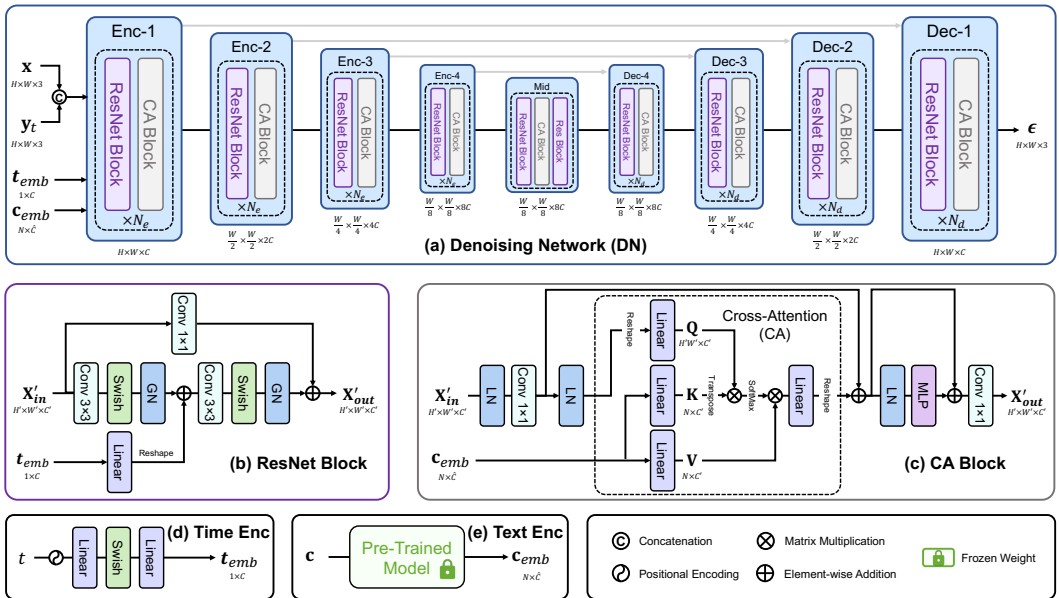

Figure 1: The detailed architecture of the PromptSR. (a) Dennoising Network (DN): an encoder-decoder U-Net architecture. Each level consists of $N_e$ (for the encoder, Enc) or $N_d$ (for the decoder, Dec) groups of ResNet and cross-attention (CA) blocks. (b) ResNet Block: comprises convolutional layers, activation layers, and group normalization (GN). Cross-Attention (CA) Block: consists of a CA module and a multi-layer perceptron (MLP). (d) Time Encoder (Enc): utilizes the positional encoding proposed in Transformer (Vaswani et al., 2017) to encode the time step $t$. (e) Text Encoder (Enc): employs a pre-trained model and freezes its weights.

## 1 PromptSR Details

We provide a detailed description of the architecture and implementation of PromptSR.

### 1.1 Architecture Detail

The architecture of the PromptSR is illustrated in Fig. 1. Given an input low-resolution (LR) image $\mathbf{x}$, it is first upsampled to the target high-resolution (HR) size $\mathbb{R}^{H \times W \times 3}$ via bicubic interpolation. Then, the LR image $\mathbf{x}$ is concatenated with the noise image $\mathbf{y}_t \in \mathbb{R}^{H \times W \times 3}$ ($t \in [1, T]$), where $t$ is the time step, and $T$ is the total step. The time step $t$ is encoded into the time embedding $\mathbf{t}_{emb} \in \mathbb{R}^{1 \times C}$ by the time encoder, where $C$ is the channel number. Concurrently, the text prompt $\mathbf{c}$ is encoded into the text embeddings $\mathbf{c}_{emb} \in \mathbb{R}^{N \times \hat{C}}$ by the text encoder, where $N$ and $\hat{C}$ are the token number and channel dimension of the text embedding. The denoising network (DN) predicts the noise $\boldsymbol{\epsilon} \in \mathbb{R}^{H \times W \times 3}$ from the LR image $\mathbf{x}$, noisy image $\mathbf{y}_t$, time embedding $\mathbf{t}_{emb}$, and text embedding $\mathbf{c}_{emb}$. The HR image $\mathbf{y} \in \mathbb{R}^{H \times W \times 3}$ is generated using the predicted noise $\boldsymbol{\epsilon}$ through several iterations.

**Cross-Attention Module.** To infuse degradation information from the text prompt into the DN, we utilize the cross-attention (CA) module. Specifically, for the input feature $\mathbf{X}_{in}^{''} \in \mathbb{R}^{H' \times W' \times C'}$ of the CA, we reshape it as $\mathbf{X}_r \in \mathbb{R}^{H'W' \times C'}$, where $H' \times W'$ denotes spatial resolution, and $C'$ is

Table 1: Model size comparisons ($\times 4$). Params (Parameters), sampler, time step, and results (PSNR/SSIM/LPIPS/DISTS) on LSDIR-Val are reported.

| Method | Params | Sampler | Time step | PSNR | SSIM | LPIPS | DISTS |
|---|---|---|---|---|---|---|---|
| Stable Diffusion (Rombach et al., 2022) | 869.12M | DDIM | 50 | 19.91 | 0.4487 | 0.4489 | 0.2240 |
| DiffBIR (Lin et al., 2024) | 1,716.71M | DDPM | 50 | 22.63 | 0.5725 | 0.3104 | 0.1758 |
| PromptSR (ours) | 215.64M | DDIM | 50 | 22.44 | 0.6070 | 0.2810 | 0.1548 |

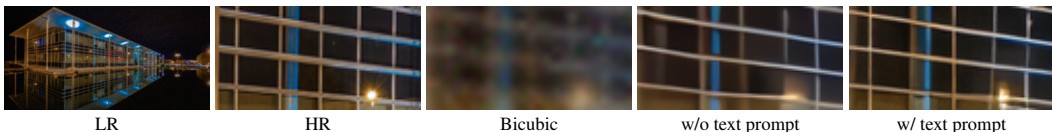

| LR | HR | Bicubic | w/o text prompt | w/ text prompt |

Figure 2: Visual comparison ($\times 4$) of text prompts. Applying (w/) the text prompt (*i.e.*, [*medium blur, upsample, light noise, light compression, downsample*]), as opposed to not using (w/o) the prompt, enables the generation of more explicit images. Please zoom in for a better view.

the channel dimension. Then, we project $\mathbf{X}_r$ as a query ($\mathbf{Q} \in \mathbb{R}^{H'W' \times C'}$) matrix. Similarly, for the text embedding $\mathbf{c}_{emb}$, encoded by the text encoder, we convert it into Key ($\mathbf{K} \in \mathbb{R}^{N \times C'}$) and Value ($\mathbf{V} \in \mathbb{R}^{N \times C'}$) matrices. The cross-attention process can be formulated as:

$$\mathbf{Q} = \mathbf{W}_Q \mathbf{X}_r, \mathbf{K} = \mathbf{W}_K \mathbf{c}_{emb}, \mathbf{V} = \mathbf{W}_V \mathbf{c}_{emb},$$
$$\mathrm{CA}(\mathbf{Q}, \mathbf{K}, \mathbf{V}) = \mathbf{W}_m(\mathrm{SoftMax}(\mathbf{Q}\mathbf{K}^T/\sqrt{C}) \cdot \mathbf{V}), \tag{1}$$

where $\mathbf{W}_Q \in \mathbb{R}^{C' \times C'}$, $\mathbf{W}_K \in \mathbb{R}^{\hat{C} \times C'}$, $\mathbf{W}_V \in \mathbb{R}^{\hat{C} \times C'}$, and $\mathbf{W}_m \in \mathbb{R}^{C' \times C'}$ are linear projections. Finally, we reshape the result of CA to obtain the output features $\mathbf{X}''_{out} \in \mathbb{R}^{H' \times W' \times C'}$. Additionally, we adopt the multi-head operation (Vaswani et al., 2017). Through CA, degradation priors from the text prompt can be integrated into the DN. The priors guide the DN to predict noise from the LR image better, thereby generating a high-quality HR image.

### 1.2 Implementation Details

The DN in our PromptSR uses a 4-level encoder-decoder architecture, with a middle layer. For the encoder, each level has $N_e$=2 sets of ResNet and CA blocks, while for the decoder, $N_d$=3. The channel dimension $C$ is set as 64. For the GN, the group number is 16. In the CA module, the number of attention heads is set as 16. For the text encoder, the token number $N$ is 77, and the channel dimension $\hat{C}$ is set as 768. Moreover, to reduce the computational complexity, we only apply CA blocks in levels 3 and 4 of the encoder and decoder, as well as in the middle layer.

## 2 Model Size Analyses

We analyze the model sizes of different diffusion-based methods, including Stable Diffusion (Rombach et al., 2022), DiffBIR (Lin et al., 2024), and our PromptSR. We report the model size (*i.e.*, Params), scheduler, timestep, and performance in Tab. 1. All metrics are calculated on the validation (Val) of LSDIR (Li et al., 2023) ($\times 4$). All models have the same time step (*i.e.*, 50). Meanwhile, for DiffBIR, we adopt the spaced DDPM sampler (Nichol & Dhariwal, 2021) as employed in the original paper, while others use the DDIM sampler (Song et al., 2020), for fairness. Compared to other methods, we can observe that our proposed PromptSR has a significantly lower parameter, accounting for only **24.8%** of Stable Diffusion and **12.6%** of DiffBIR. Meanwhile, our proposed PromptSR outperforms other diffusion methods on most metrics. For instance, on the LPIPS, our approach achieves a reduction of 0.0294 compared to DiffBIR.

## 3 More Visualizations on Text Prompt

We provide more visual comparisons related to the text prompt.

LR     [*unchange*] (✗)     [*downsample*] (✓)     LR     [*heavy noise*] (✗)     [*light noise*] (✓)

Figure 3: Visual comparison (×4) of different prompts. Template prompt: [***light blur, unchange, light noise, light compression, downsample***], with underlined parts indicating substitutions. Left: incorrect description vs. proper description. Right: over description vs. proper description.

## 3.1 TEXT PROMPT

We present the reconstruction results with (w/) and without (w/o) text prompts in Fig. 2. Text prompts lead to clearer visual results. This improvement is attributed to the text prompt providing a degradation prior, enabling better modeling of degradation.

## 3.2 DIFFERENT PROMPTS

We compare the recovery results using different prompts in Fig. 3 and also provide the complete template prompt (also used in Fig. 5 of the main paper). It is evident that suitable prompts can yield higher-quality results. For the image on the right, using the prompt [***heavy noise***] effectively removes most of the noise but also leads to over-smoothing and loss of details. In contrast, reducing the noise level specified in the prompt helps recover more realistic results. These findings further demonstrate the effectiveness and flexibility of text prompts.

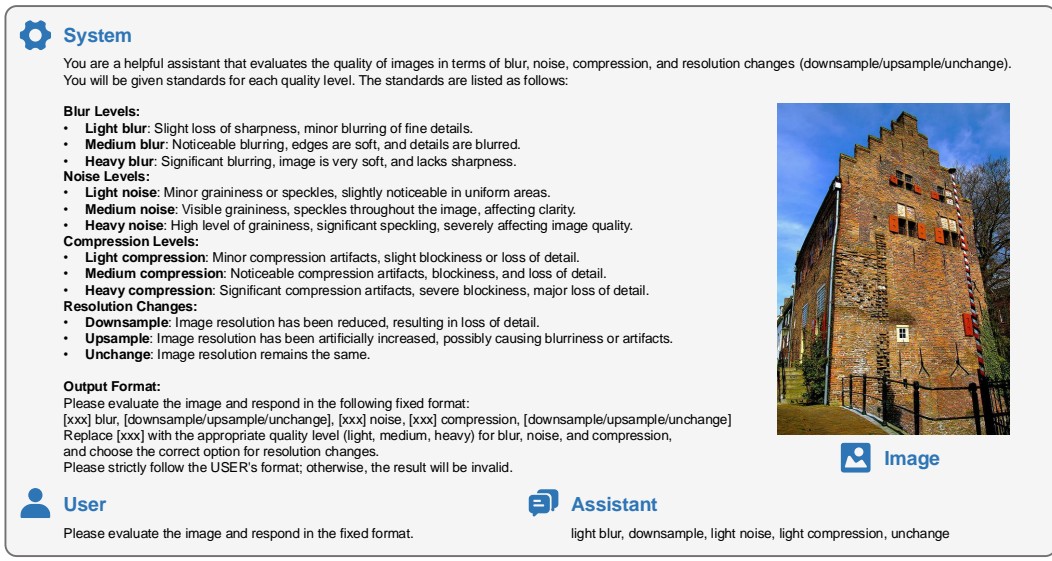

Figure 4: Example of prompt generation using MLLM. It includes (real-world) image, system message, user message, and assistant response. The assistant response represents the generated prompt.

## 4 MORE DETAILS OF PROMPT IN REAL

In this section, we provide more detailed information on applying our designed prompts to real-world images. We first introduce the specific configurations for generating prompts using multi-modal large language models (MLLMs) (Liu et al., 2023; OpenAI, 2023; Ye et al., 2024; Wu et al., 2024). Then, we provide more examples of prompts corresponding to real images from Real45.

Table 2: More text prompts for real-world instances. The low-quality images are also provided.

| Image | Prompt | Image | Prompt |
|---|---|---|---|
|  | *light blur, downsample, medium noise, medium compression, downsample* |  | *light blur, downsample, light noise, light compression, downsample* |

### 4.1 MLLM GENERATION DETAILS

We employ the advanced open-source multi-modal large language model, mPLUG-Owl3 (Ye et al., 2024) (specifically, ***mPLUG-Owl3-7B***), to generate prompts from given real-world images. Detailed prompt settings are provided in Fig. 4. Furthermore, we **optionally** fine-tune the MLLM-generated prompts 1-2 times based on the restoration results.

### 4.2 MORE TEXT PROMPT DETAILS

In Tab. 2, we present more prompts corresponding to real-world images. Through the prompts, the method can better model the degradation, thereby reconstructing more realistic and clear images. Complete text prompt files will be released alongside the code.

## 5 MORE VISUAL RESULTS

In Figs. 5 and 6, we provide more visual comparisons on both synthetic and real-world datasets. Our proposed PromptSR, compared to other methods, handles challenging cases more effectively and recovers images with more details. For instance, in the synthetic dataset, the first example of Urban100, our model can restore sharper textures (lines), whereas other comparison methods introduce undesirable artifacts. In the real-world dataset, the first instance of Real45, the recovery results from other methods are inconsistent with reality. In contrast, our PromptSR can restore more realistic and faithful outcomes. These results, supplementing the main paper, further demonstrate the superiority of introducing text prompts into image SR.

## 6 LIMITATIONS AND FUTURE WORK

In this work, we introduce text prompts into image SR to provide degradation priors. Our prompts are simple, flexible, and user-friendly. However, our prompts employ the tag-style format, which diverges from natural expression. Furthermore, to simplify the representation, we utilize three binning categories (*i.e.*, 'light', 'medium', and 'heavy'), leading to a coarse control granularity.

In future work, we plan to explore text prompt forms that align with natural expressions. This requires the model to extract key information from redundant descriptions for restoration. Additionally, while general descriptions of the overall image content are not significantly beneficial for image SR, more specific descriptions of details (such as facial features) may be effective.

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

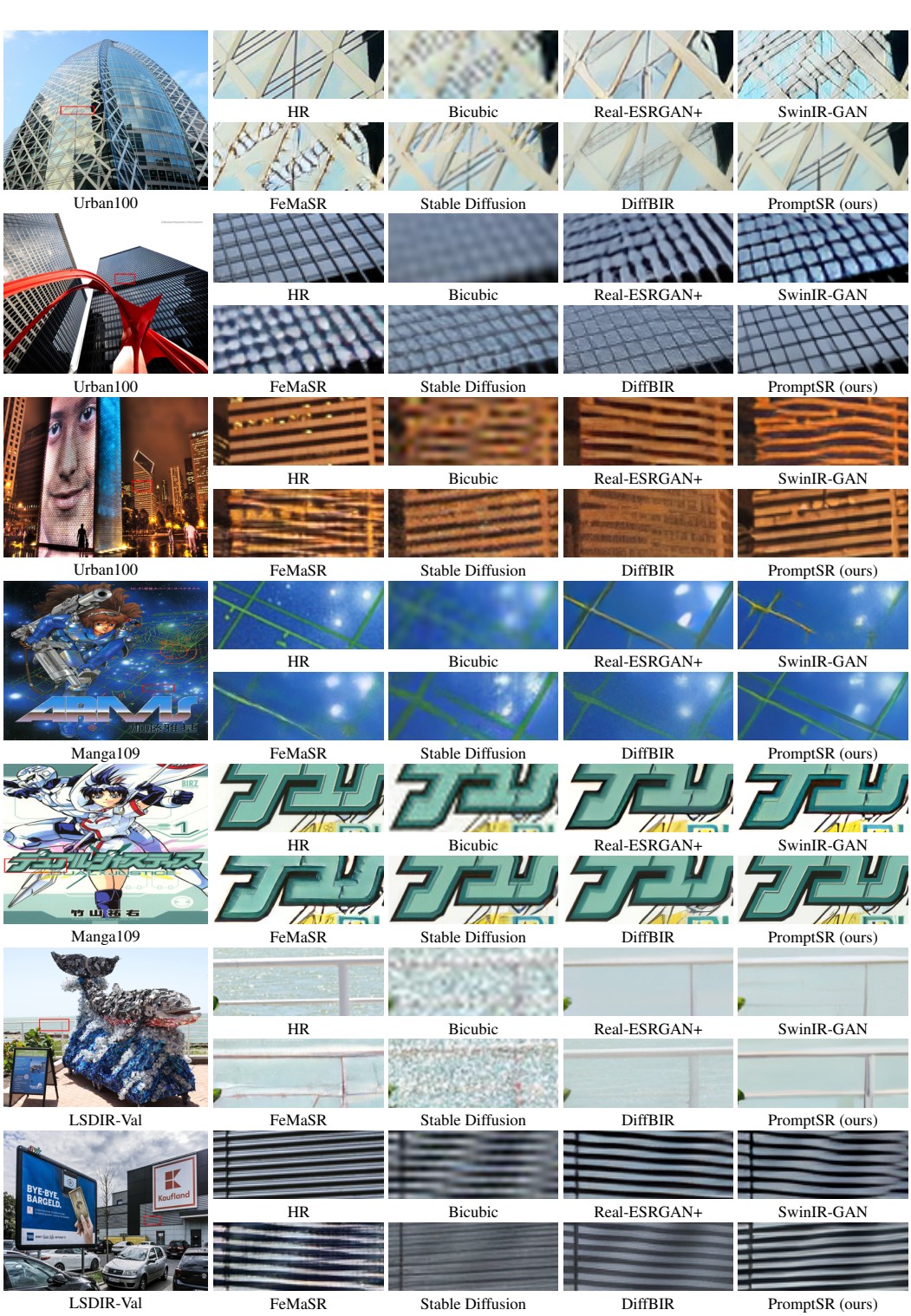

Figure 5: Visual comparison (×4) on Urban100 (Huang et al., 2015), Manga109 (Matsui et al., 2017), and LSDIR-Val (Li et al., 2023) datasets. Please zoom in for a better view.

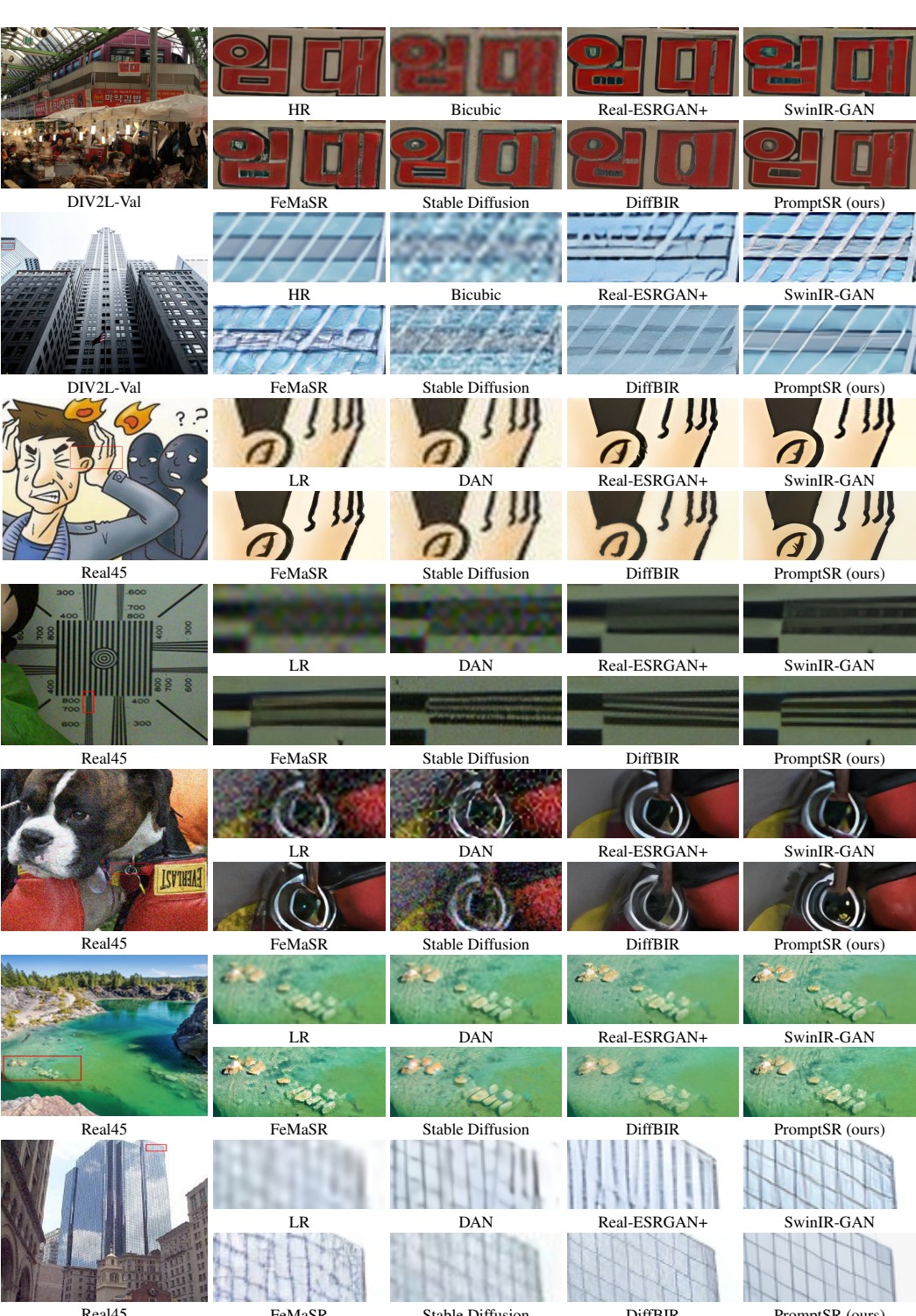

Figure 6: Visual comparison (×4) on DIV2K-Val (Timofte et al., 2017) and Real45 datasets (collected from the internet). Please zoom in for a better view.