# OpenReview forum: "Image Super-Resolution with Text Prompt Diffusion"
_ICLR.cc/2025/Conference — ICLR 2025 Conference Withdrawn Submission_

### Official Review · Reviewer_iDK6 · 2024-10-29

**Soundness:** 2
**Presentation:** 2
**Contribution:** 2
**Rating:** 5
**Confidence:** 5

**Summary:**

The work introduces text prompts to image SR to provide degradation priors and develops a text-image generation pipeline to integrate text into the SR dataset.

**Strengths:**

1. The work develops a text-image generation pipeline that integrates prompt into the SR dataset via text representation and degradation model.
2. The work proposes PromptSR, which utilizes the pre-trained language model to improve the restoration.
3. Experiments show the effectiveness of the proposed method.

**Weaknesses:**

1. The work lacks comparison with state-of-the-art methods [1,2,3].
2. The work should conduct experiments on more real-world datasets, e.g. DRealSR dataset.
3. The work does not show how user-friendly and flexible the prompt is. To some extent, it is also flexible to directly give the user a 0-1 value as the strength of each degradation.
4. Using a text encoder to encode discrete degradations is somewhat redundant. Does the method still work when the degradation description is changed (e.g., heavy blur -> very blurry)?


[1] Scaling Up to Excellence:Practicing Model Scaling for Photo-Realistic Image Restoration In the Wild. CVPR 2024.
[2] SeeSR: Towards Semantics-Aware Real-World Image Super-Resolution. CVPR 2024.
[3] CoSeR: Bridging Image and Language for Cognitive Super-Resolution. CVPR 2024.

**Questions:**

None.

---

### Official Review · Reviewer_JpUv · 2024-11-01

**Soundness:** 2
**Presentation:** 2
**Contribution:** 2
**Rating:** 3
**Confidence:** 5

**Summary:**

This paper introduces textual prompt information into the image super-resolution (ISR) task to provide degradation priors. It proposes a text-image generation pipeline that integrates text prompts into the SR dataset. The proposed method, PromptSR, leverages pre-trained language models to facilitate image restoration.

**Strengths:**

- The paper is well-structured.
- It explores the effective role of textual information in the super-resolution task.

**Weaknesses:**

1. **Limited Novelty**: The idea of degradation-guided RealSR has been extensively explored in numerous low-level vision papers, including but not limited to:
   - *Efficient and Degradation-Adaptive Network for Real-World Image Super-Resolution (DASR)*
   - *Textual Prompt Guided Image Restoration*
   - *Dcs-risr: Dynamic Channel Splitting for Efficient Real-World Image Super-Resolution*
   - *DaLPSR: Leverage Degradation-Aligned Language Prompt for Real-World Image Super-Resolution*

2.  **What is the necessity of the text?**: The proposed integration of degradation information into the SR network is not significantly different from the approach used in DASR. The text used in this paper, aside from providing degradation classification information, does not offer any additional value. Thus, the necessity of integrating text and a text encoder into the SR network is questionable. It is possible that using DASR’s degradation features could achieve similar results.

3. **Lacks comparisons with many popular SR methods**:  The paper almost entirely omits comparisons with diffusion-based SR methods. There are many open-source and popular diffusion-based methods, including but not limited to:
   - *[IJCV2024] Exploiting Diffusion Prior for Real-World Image Super-Resolution*
   - *[CVPR2024] SeeSR: Towards Semantics-Aware Real-World Image Super-Resolution*
   - *[ECCV2024] Pixel-Aware Stable Diffusion for Realistic Image Super-Resolution and Personalized Stylization*
   - *[CVPR2024] CoSeR: Bridging Image and Language for Cognitive Super-Resolution*
   - *[CVPR2024] Scaling Up to Excellence: Practicing Model Scaling for Photo-Realistic Image Restoration In the Wild*
   - *[CVPR 2024] SinSR: Diffusion-Based Image Super-Resolution in a Single Step*
This makes it difficult to be convinced of the proposed method’s superiority.

This paper appears to be outdated, as the field of RealSR has advanced rapidly. I don’t believe this paper contributes value to the field. Considering the limited novelty and the insufficient experiments,  I decided to give it a rejection.

**Questions:**

As shown in Weaknesses

---

### Official Review · Reviewer_PFXi · 2024-11-02

**Soundness:** 2
**Presentation:** 2
**Contribution:** 1
**Rating:** 3
**Confidence:** 4

**Summary:**

This paper proposes to use textual prompts in image super-resolution tasks in the following way: they first design a training data generation pipeline by degrading the original high resolution image in a sequence of chosen transformation steps (blur, upsample, noise, compression, downsample), and in parallel they construct a textual prompt which describes the applied transformations on the original sample. Then they design a U-Net like denoiser network with cross-attention layers to be used as a noise predictor in the diffusion model framework. The inputs to this network comprise of the noised version of the original resolution image patch concatenated with the corresponding low resolution image patch resized to the original image patch size. They use a text encoder model (such as CLIP or T5) to embed the constructed textual prompt as a sequence of vectors and use it as a prior for the diffusion model guidance. Experimental results show the effectiveness of the proposed method.

**Strengths:**

The proposed method is well described.

**Weaknesses:**

1. The use of textual prompts in image super-resolution tasks is not novel, yet the paper lacks discussion and comparison with existing methods like PASD [1] and SeeSR [2], which also employ textual prompts.

2. In Section 3.1.2, the paper claims that textual prompts depicting degradation are superior to prompts based on image content for conditioning the denoiser network, referencing Figure 3 as evidence. However, it does not clarify how the "overall caption" result was generated, so further explanation is needed. Additionally, a comprehensive analysis and comparison with related works [1, 2] is necessary rather than relying on a single example to assert that semantic textual descriptions are redundant.

3. The paper suggests using a pre-trained MLLM for generating the degradation description for real-world super-resolution inference. But it doesn’t analyze how often the MLLM-generated prompts match the true data degradation procedure. It’s not clear why the paper assumes that the MLLM can give a good description about the image degradation in real-world use-cases.  It would be beneficial to report the accuracy of the MLLM-generated prompts.

4. Given that BSRGAN [3] outperforms or closely matches the proposed method on some metrics in Table 7, the paper needs to also include BSRGAN in qualitative results.

5. The second comparison in Figure 7 includes the DAN result, while the first comparison lacks. The paper needs to make a proper qualitative comparison with all methods.

6. The paper lacks a user study both on synthetic and real-world datasets.

References

[1] Yang, Tao, Rongyuan Wu, Peiran Ren, Xuansong Xie, and Lei Zhang. "Pixel-aware stable diffusion for realistic image super-resolution and personalized stylization." arXiv preprint arXiv:2308.14469 (2023).

[2] Wu, Rongyuan, Tao Yang, Lingchen Sun, Zhengqiang Zhang, Shuai Li, and Lei Zhang. "Seesr: Towards semantics-aware real-world image super-resolution." In Proceedings of the IEEE/CVF conference on computer vision and pattern recognition, pp. 25456-25467. 2024.

[3] Zhang, Kai, Jingyun Liang, Luc Van Gool, and Radu Timofte. "Designing a practical degradation model for deep blind image super-resolution." In Proceedings of the IEEE/CVF International Conference on Computer Vision, pp. 4791-4800. 2021.

[4] Huang, Yan, Shang Li, Liang Wang, and Tieniu Tan. "Unfolding the alternating optimization for blind super resolution." Advances in Neural Information Processing Systems 33 (2020): 5632-5643.

**Questions:**

Given that the textual prompts are limited to a small set of predefined options—light/medium/heavy blur, light/medium/heavy noise, light/medium/heavy compression, upsample, and downsample—why not replace the text encoder with a set of learnable embeddings? Seems using just 11 learnable embeddings could capture each transformation eliminating the need for a text encoder.

---

### Official Review · Reviewer_JbNu · 2024-11-03

**Soundness:** 3
**Presentation:** 3
**Contribution:** 3
**Rating:** 6
**Confidence:** 5

**Summary:**

This paper introduces the text prompts to provide degradation priors for enhancing image SR. Specifically, the authors first develop a text-image generation pipeline to integrate text into the SR dataset, via text degradation representation and degradation model. Then, they further propose the PromptSR to realize the text prompt SR. The PromptSR applies the pre-trained language model to enhance text guidance and improve performance. Extensive experiments on both synthetic and real-world datasets demonstrate the effectiveness of introducing text into SR.

**Strengths:**

1, The proposed prompt text includes some degradation, such as Blur, Resize, Noise, Compression. The reviewer wonders to know why contains the scale factor? Is it flexible to embed the scale factor to SR model?

2, This paper introduces a new pipline for how to measure the degradation which serves as prior to effectively guide deep models.

3, The authors varify the effectiveness of the proposed PromptSR compared with existing generative-based models, including FeMaSR, DiffBIR, etc.

4, The analysis is adequate.

**Weaknesses:**

1, The proposed prompt text includes some degradation, such as Blur, Resize, Noise, Compression. The reviewer wonders to know why contains the scale factor?  Is it flexible to embed the scale factor to SR model?

2, The reviewer would like to know the inference time.

3, Do the authors consider the our-of-distribution case when inference? For example, the testing image contains blur and noise, but the text prompt when inference only has 3 text prompts, e.g. blur, noise, compression.

4, Do the authors consider the our-of-distribution case when training? For example, the training text contains blur and noise, but the testing image contains 3 degradations, e.g., blur, noise, and compression.

**Questions:**

See Weaknesses

---

### Note · Authors · 2024-11-15

I have read and agree with the venue's withdrawal policy on behalf of myself and my co-authors.